# Phytosanitary Rules for the Movement of Olive (*Olea europaea* L.) Propagation Material into the European Union (EU)

**DOI:** 10.3390/plants12040699

**Published:** 2023-02-04

**Authors:** Vito Montilon, Oriana Potere, Leonardo Susca, Giovanna Bottalico

**Affiliations:** Department of Soil, Plant and Food Sciences (DiSSPA), University of Bari Aldo Moro, 70126 Bari, Italy

**Keywords:** olive tree, germplasm conservation, phytosanitary legislation, pest management, quarantine pest, priority pests, Union regulated nonquarantine pests, plant passport, phytosanitary certificate

## Abstract

Phytosanitary legislation involves government laws that are essential to minimize the risk of the introduction and diffusion of pests, especially invasive non-native species, as a consequence of the international exchange of plant material, thus allowing us to safeguard agricultural production and biodiversity of a territory. These measures ensure compliance with adequate requirements relating to the absence of pests, especially of harmful quarantine organisms through inspections and diagnosis tests of the consignments to ascertain the presence of the pests concerned. They also regulate the eradication and containment measures that are implemented in the eventuality of an unintentional introduction of these organisms. In the present contribution, the current plant protection legislation for the exchange of plants or propagation material within the European Union or for export to foreign countries, represented by Regulation (EU) 2016/2031, has been reviewed, with a particular focus on the olive tree (*Olea europaea* L.). Furthermore, a brief summary of the main olive tree pests transmissible with the propagation material is also reported, indicating their current categorization with respect to the relative quarantine status.

## 1. Introduction

Outbreaks due to non-native invasive pests have become more frequent because of the increased movement of plants and agricultural products around the world, which takes place for commercial or research purposes. The introduction of these harmful organisms can cause serious damage to the agricultural production of a territory, representing one of the causes responsible for the genetic erosion of many plant species [1]. In reference to this, a recent example is the epidemic of olive quick disease syndrome (OQDS) caused by the bacterium *Xylella fastidiosa* that is native to the American continent, which was detected in 2013 in South Italy [2,3]. Prevention of the introduction of these harmful organisms is primarily accomplished through international quarantine regulations and adequate controls on transboundary shipments of plants/propagation material. Plant health regulations allow agents to exchange only plant material that meets specific phytosanitary requirements, particularly as regards their freedom from specific pests [4]. Verification of these requirements is achieved by inspecting batches of plants or propagation material through visual surveys for the presence of symptoms, and by means of diagnosis through laboratory analysis using serological or molecular tests. The policies of different countries regarding plant quarantines are in line with the International Convention for the Protection of Plants (IPPC) [5]. This convention is a multilateral treaty that currently includes 183 countries, which was signed on 6 December 1951 under the aegis of the Food and Agriculture Organization of the United Nations (FAO) and later revised in November 1997 [6]. It aims to coordinate international policies for the prevention of the introduction and spread of harmful organisms of plants, with the term “plants” meaning living plants and parts of them, seeds and germplasm [7]. The governing body of the IPPC is constituted by the Commission on Phytosanitary Measures (CPM) while the IPPC’s secretariat defines the recommended measures for plant protection constituted by the International Standards for Phytosanitary Measures (ISPMs) (https://www.ippc.int/core-activities/standards-setting/ispms, accessed on 19 October 2022). According to the IPPC, a “pest” is defined as “any species, strain or bio-type of plant, animal or pathogenic agent injurious to plants or plant products”, while a quarantine pest is a “pest of potential economic importance to an area and not yet present, or if present is not widely distributed and subject to official controls” [8]. The IPPC guidelines provide for a hierarchical approach that considers the prevention of the unintended entry of plant pests as a priority over the eradication and subsequent containment interventions [9]. The International Plant Protection Convention (IPPC) standards are also recognized by the World Trade Organization’s (WTO) through the Agreement on the Application of Sanitary and Phytosanitary Measures (SPS) [10]. The ISPMs of the IPPC constitute a reference for the Regional Organizations for the Protection of Plants (RPPOs), which have the role of harmonizing the phytosanitary measures between the National Plant Protection Organizations (NPPOs). NPPOs are services authorized by government authorities for the implementation of the guidelines issued by the IPPC. Regulatory measures can be established by the states based on the recommendations of these plant health organizations. The RPPO for the Euro-Mediterranean region is constituted by the European and Mediterranean Plant Protection Organization (EPPO), which was founded in 1951. The EPPO, in addition to coordinating IPPC guidelines between the member NPPOs, defines plant protection standards which can be used by the NPPOs of member states to establish laws on the prevention of the introduction of harmful organisms or to limit their spread in the case of entry. The EPPO also draws up alert lists, which are lists of pests that are potentially harmful to the Mediterranean area and that can be considered as quarantine pests. Periodically, following a process of pest risk analysis (PRAs), some of these listed organisms can be inserted on the EPPO A1 and A2 lists or removed from them if the relative risk is no longer judged to be high. These EPPO A1 and A2 lists include pests recommended to be regulated as quarantine pests. The EPPO A1 list includes exotic quarantine pests which are absent from the EPPO region; instead, the A2 list reports pests whose presence has already been reported in the territory of the EPPO region. In the latter list, the pathogen *X. fastidiosa* was included in 2017 following its discovery in Europe, which took place initially in southern Italy, which was followed by its detection in other areas in Europe [11]. Another function of the EPPO is to standardize the phytosanitary diagnostic procedures in the EPPO region, which is accomplished through the approval of diagnostic standards on plant pests and especially for quarantine pests. EPPO Diagnostic Standards are published in the EPPO Bulletin and uploaded to the EPPO database. They include both horizontal standards covering quality assurance on performing diagnostic tests, and specific diagnostic standards consisting of protocols for detecting specific pests. Diagnostic standards have also been established for the detection of *X. fastidiosa* [12]. The EPPO also manages the EPPO Global Database [13], which is a constantly updated database containing the EPPO’s standards and information on plant pest species or invasive alien plants, also providing the host plants and quarantine status for each pest. 

Another European reference authority for the protection of plants from harmful organisms is the European Food Safety Authority (EFSA). The EFSA is an agency of the European Union established in 2002 with the function of supporting the decisions of the European institutions and governments concerning the protection of consumer health and food safety. One of the EFSA’s mandates is to participate in the protection of EU member states from the threat posed by plant pests. In this regard, the EFSA draws up pest survey cards containing pest-specific information and guidelines based on international standards to assist EU countries in plant pest surveys. The European Community’s phytosanitary regime complies with the principles of the IPPC and of the Food and Agriculture Organization of the United Nations (FAO). In consideration of the importance of phytosanitary policies to minimize the risk of spreading harmful organisms to plants, the phytosanitary legislation in force in the European community was reviewed in this work, with particular focus on the olive tree (*Olea europaea* L. subsp. *europaea*). Furthermore, the olive tree pests transmissible by propagation material and their relative current categorization in the European Union were also described.

## 2. Olive Tree Pests Transmissible by Propagation Material

Harmful agents of the olive tree include fungi, phytoplasm, viruses and virus-like agents, bacteria and nematodes, which can spread even over a long distance with the movement of the infected propagation material [14] and may be responsible for negative economic effects on the production of this crop [15]. Among the fungal pests, *Verticillium dahliae* Kleb. [16] is a vascular, soil-inhabiting pathogen responsible for verticillium wilt, which represents one of the main diseases of the olive tree [15]. The species of phytoplasma that affect the olive tree can cause alterations such as reduced growth, “witches’ broom” phenomena, and deformation and yellowing of the leaves. They include *Candidatus Phytoplasma asteris*, *Candidatus Phytoplasma solani*, *Candidatus Phytoplasma ulmi* and *Candidatus Phytoplasma pruni* [17,18]. The most important olive viruses are *Olive leaf yellowing associated virus* (OLYaV) [19], *Olive yellow mottling and decline-associated virus* (OYMDaV) [20], *Olive latent virus-1* (OLV-1) [21], *Olive latent virus-2* (OLV-2) [22], *Olive latent virus 3* (OLV-3) [23], *Tobacco mosaic virus* (TMV) [24], *Olive vein yellowing-associated virus* (OVYaV) [25], *Olive mild mosaic virus* (OMMV) [26], *Arabis mosaic virus* (ArMV) [27], *Cherry leaf roll virus* (CLRV) [28], the soil-borne virus *Strawberry latent ring spot virus* (SLRSV) [27], *Cucumber mosaic virus* (CMV) [29], *Tobacco necrosis virus* Strain D (TNV-D) [30], *Olive semilatent virus* (OSLV) [31] and *Olive latent ringspot virus* (OLRSV) [32]. SLRSV, ArMV and TNV are soil-borne; CLRV and OLV-1 can be transmitted by the seeds; CMV is transmitted by aphids; and OLV-2 and OLRSV are transmitted both mechanically and by grafting [33]. Furthermore, two new olive-infecting viruses have recently been discovered, the Olea europaea geminivirus (OEGV) [34] and the olive virus T (OlVT) [35]. In olive trees, viral infections are often asymptomatic, and some viruses can lead to symptoms only in some cultivars while remaining latent in others, as in the case of SLRSV, which is the etiological agent of the infection that has the name of bumpy fruit [36,37]. The development of symptoms also occurs due to infections of the OVYaV, OYMDaV and OLYaV viruses, which are responsible for leaf yellowing complex [37], and by strains of TMV and OSLV isolated from olive trees with symptoms of vein banding and vein clearing diseases, respectively [15]. Moreover, in some cases, olive tree viruses can determine a reduction in growth and rhizogenic capacity [37]. Although these viruses can remain latent or cause negligible damage, some of them can severely affect other plant species; in particular, SLRSV can cause Black line disease in walnut [33]. The olive tree is also affected by some virus-like diseases of unknown etiology, for some of which, transmission by graft has been successfully achieved. These include partial paralysis, foliar deformation, sickle leaf, infectious yellowing, spherosis and bark cracking [38,39]. Among the bacteria, *Pseudomonas savastanoi* pv. *savastanoi* (E.E. Smith) is a pathogen that causes a widely diffused alteration in olive knot disease [15]. In addition, the phytopathogenic bacterium *Xylella fastidiosa* [40] is present in Southern Italy, where a new isolate of *X. fastidiosa* subsp. *pauca*, classified as the ST53 sequence type [41,42], was found to be associated with olive quick disease syndrome (OQDS) [3]. Infections caused by *X. fastidiosa* on olive trees have also been reported in Argentina and Brazil, which were caused by two different sequence types of *X. fastidiosa* subsp. *pauca*, consisting, respectively, of ST69 [43,44], and ST16 [45], and have also been reported in California due to *X. fastidiosa* subsp. *multiplex* [46]. The olive tree is also affected by several species of nematodes including the species *Meloidogyne arenaria* Chitwood [47], *Meloidogyne incognita* (Kofoid & White) Chitwood [48], *Meloidogyne javanica* Chitwood [49], *Pratylenchus vulnus* [50] and *Xiphinema diversicaudatum* [51]. Among these, *X. diversicaudatum* is also responsible for the transmission of ArMV and SLRSV [52], while *M. incognita* and *P. vulnus* favor the penetration of *Verticillium dahliae*, causing injuries to the plants’ roots [52].

## 3. European Regulation (EU) 2016/2031 on the Protective Measures against Plant Pests

In the European Community (EU), the current legal framework on plant protection is constituted by Regulation (EU) 2016/2031 of 26 October 2016 [53], and the related delegated or implementing regulations [54,55,56,57,58,59,60,61,62] (Table 1). This regulation entered into force on 13 December 2016 and constitutes a thorough revision of the previous directive, Council Directive 2000/29/ EC [63]. In this section, some of the general items in the reference articles that are envisaged by the EU regulation are reported. The regulation provides for the establishment of lists of Union-regulated nonquarantine pests, Union quarantine pests and priority pests by the member states. These lists are subject to continual updating as a result of an ongoing re-evaluation process of the risk posed by the various harmful organisms on the basis of recent technical and scientific developments.

A quarantine pest (Article 3) is defined a pest which is not present in a given territory where it is capable of becoming established or, if already present, is not widely distributed and which would have an unacceptable economic impact on that territory. Union quarantine pests (Article 4) are quarantine pests for the European Union territory, whose introduction or movement is prohibited in the EU, and include pests that are listed in Annex I, Part A of Directive 2000/29/EC [63] and in Annex II, Part A, Section I, of this directive. They may be organisms known to occur in the territory of the Community that are either indigenous or established as a result of their introduction from outside, as well as exotic pests not known to occur in the Community’s territory. The list of Union quarantine pests is reported in Annex II of the Commission Implementing Regulation (EU) 2019/2072 of 28 November 2019 [58]. In particular, Annex II is divided into two parts. Part A includes pests whose presence is not known in the Union’s territory, while Part B includes pests whose presence has been ascertained. Priority pests (Article 6) are quarantine harmful organisms whose potential economic, environmental or social impact is the most severe in respect of the Union’s territory, and for which a strengthening of the surveillance investigations is envisaged. The list of priority pests is reported in the Commission Delegated Regulation (EU) 2019/1702 of 1 August 2019 [59]. Union-regulated non-quarantine pests (Article 36) are organisms widely spread in the EU which are mainly transmitted through various plants for planting and whose presence in these crop plants has a negative economic effect. Given their widespread presence in the Community’s territory, they no longer meet the criteria of a quarantine organism, although measures must still be taken to prevent their presence in plant propagation material. For these reasons, professional operators, consisting of those who introduce or move plants and plant products within the Union, cannot commercialize propagation material affected by this category of pests within the Community’s territory. The list of Union-regulated nonquarantine pests is reported in Annex IV of the Commission Implementing Regulation (EU) 2019/2072 [58]. 

In line with the previous legislation on plant protection (Directive 2000/29/EC), Regulation (EU) 2016/2031 provides that plants or plant products must be accompanied by a plant passport for crossing borders between EU member states (Article 78). A plant passport certifies the compliance of the plant material with the established phytosanitary requirements and, in particular, the absence of Union quarantine pests and compliance with the prevention measures for Union-regulated nonquarantine pests. In addition, it must ensure that the plants are free from pests which are not included in the list of quarantine pests but which the Commission considers may satisfy the conditions for inclusion in that list (Article 30). The format of a plant passport is specified by the Commission Implementing Regulation (EU) 2017/2313 of 13 December 2017 [54]. A plant passport is issued by the professional operators registered in the official register of professional operators (Article 65) and authorized by competent authorities represented by the phytosanitary services (Article 89). The criteria that must be met by professional operators to be authorized to issue passports are reported in the Commission Delegated Regulation (EU) 2019/827 of 13 March 2019 [57]. For the introduction of plants or plant products from foreign countries into the European Union, a phytosanitary certificate is required (Article 71). It certifies that the plants/plant products are free from Union quarantine pests and their compliance with the preventive measures for Union-regulated nonquarantine pests as well as pests that are not included in the list of quarantine pests but which the Commission considers could meet the conditions for inclusion in that list (Article 30). This certificate is issued by a foreign country if the requirements set out in Article 76 are met and its format is reported in Part A of Annex V. The phytosanitary certificate and the plant passport are mandatory for all plants used for planting. Furthermore, a phytosanitary certificate and a plant passport are required for the introduction of plants or plant products from foreign countries (Article 74) or from a member country of the European Union (Article 86) into a protected area. A protected area is a territory of a member state or part of it that is free from a quarantine pest present within the Union, although favorable environmental conditions exist for its establishment. For this reason, a prohibition or a restriction is foreseen regarding the import of certain plants from foreign countries or other EU countries within these areas (Article 32). 

For the export or re-export of plants from the EU to a foreign country, it is necessary to use a phytosanitary certificate for export and re-export, and a pre-export certificate (Article 100). These certificates are issued by competent authorities at the request of the registered professional operators if the necessary requirements are met. A phytosanitary certificate for export from the Union (Article 100) is necessary for exporting a plant from the Union to a foreign country, ensuring compliance with the phytosanitary requirements of the importing country, and a model for this is presented in Part A of Annex VIII. A phytosanitary certificate for re-export from the Union (Article 101) is necessary for re-exporting a plant originating from a foreign country that has been introduced in the EU territory to another country. This certificate guarantees that plants are free from the pathogens established by the foreign importer, and the format for this is reported in Part B of Annex VIII. A pre-export certificate (Article 102) allows the member state in which the plants were grown or produced to guarantee their phytosanitary quality to a member state that exports these materials abroad. It accompanies the plant material in its movement within the Union and the format for this certificate is reported in Part C of Annex VIII. Figure 1 shows the key steps and phytosanitary certificates for importing or exporting propagation materials into and out of the European Union (EU).

According to the current legislation, the Commission should provide for an assessment of the risk of introducing a pest on the basis of the scientific opinions of the IPPC, the EPPO, EFSA and the member states’ authorities. The regulation also defines the planning of surveys that must be carried out to protect against the introduction of Union quarantine pests (Article 22) and of priority pests (Article 24) into the Community. For priority pests, it is also envisaged that member states must develop contingency and action plans for the eventuality of accidental introductions, in addition to simulations related to these intervention plans (Article 26) and the diagnostic procedures to be followed. In this case, the member states must report the detection of quarantine pests to the Commission, either in the case of an occurrence in the Community’s territory or in a batch of imported plant material. In case of a confirmed detection of a Union quarantine pest, the competent authority must take measures to eradicate that organism from the affected area (Article 17). As part of these activities, a phytopathological diagnosis represents an integral part of the plant quarantine, allowing the identification and characterization of the strains/isolates of pests. This task is entrusted to laboratories officially recognized by the competent authorities. Measures for the eradication of quarantine pathogens include the establishment of demarcated areas, consisting of an infested zone and a buffer zone (Article 18), and monitoring of the demarcated area to detect these pathogens (Article 19). A derogation has been established for certain provisions of Regulation (EU) 2016/2031 for regulated pests or plants which can be introduced or moved within the EU or in protected areas for official tests, scientific or educational purposes, experiments, and/or variety selection or breeding, in accordance with the Commission Delegated Regulation (EU) 2019/829 [60].

## 4. Pest Categorization of the Main Harmful Agents of the Olive Tree

The legislation currently in force for the exchange of olive trees or propagation material within the European Union or for export to foreign countries is represented by Regulation (EU) 2016/2031 [53]. It defines the phytosanitary requirements that must be met for this plant species to prevent the introduction and spread of harmful organisms. With reference to this regulation, there are 13 olive tree pathogens that meet the criteria to be specified as Union-regulated non-quarantine pests, while a single pathogen consisting of a bacterium is regulated as a Union priority pest. Specifically, the Union-regulated nonquarantine pests of olive include the viruses SLRSV, ArMV, CLRV, OLYaV, OVYaV and OYMDaV; the nematodes *Meloidogyne arenaria* Chitwood, *Meloidogyne incognita* (Kofoid & White) Chitwood, *Meloidogyne javanica* Chitwood and *Pratylenchus vulnus*, *Xiphinema diversicaudatum*; the bacterium *Pseudomonas savastanoi* pv. *savastanoi* (E.E. Smith); and the fungal pathogen *Verticillium dahliae* Kleb. These pathogens are listed in Part J of Annex IV, of the Commission Implementing Regulation (EU) 2019/2072 concerning the propagation material of fruit plants for planting [58]. Among the olive tree pests, the pathogen *Xylella fastidiosa* is regulated as a priority quarantine pest in the EU [59]. This bacterium is included in the list of the Commission Delegated Regulation (EU) 2019/1702 of 1 August 2019 [59]. Furthermore, it is also included in the Part B of Annex II of the Commission Implementing Regulation (EU) 2019/2072 comprising the Union quarantine pests known to occur in the Community’s territory [58]. In the European Union, the interventions to be adopted for the prevention of *X. fastidiosa* and the measures for its control and eradication in areas where there are outbreaks of this pathogen are regulated by the Commission Implementing Regulation (EU) 2020/1201 of 14 August 2020 [62]. This regulation provides that the member states must carry out annual surveys of the territory through visual inspections for disease symptoms and diagnostic tests for the identification of the pathogen, and also by identification of the subspecies in the event of its detection. The detection of the pathogen and identification of the subspecies must take place through molecular analysis following the procedures indicated in the annex of the regulation, and must be carried out by competent authorities or in laboratories authorized by the competent authorities. This regulation also requires that member states must establish a contingency plan in case the presence of the bacterium is confirmed. This plan involves establishing demarcated areas consisting of an infected zone and a buffer zone, and applying eradication and containment measures. In the delimited areas, in addition to surveys of the host plant species for the presence of the bacterium, monitoring and control of the vector insects must also be carried out. The movement of plants/propagation material out from the delimited areas and within the infected areas is regulated, as well as the introduction of host plants into the EU and the planting of host species in infected areas. Furthermore, in order to be imported into Europe, plants originating from foreign countries where a harmful agent is not present must be accompanied by a phytosanitary certificate containing a further declaration specifying that the country of origin is free from the harmful organism in question.

## 5. EPPO Standard PM 4/17 (3) on the Certification Scheme for the Production of Olive Tree Varieties and Rootstocks

The EPPO Standard PM 4/17 (3) concerns the guidelines on the production of pathogen-tested olive tree varieties or rootstocks for planting [64]. This standard states that the certification scheme should start from the source plants, called nuclear stocks, which are propagated through various phases in order to control the phytosanitary status and the trueness to type of the plants produced, and also guaranteeing the traceability of the entire supply chain [65]. Nuclear stocks are obtained by selection for the agronomic characteristics of interest and the absence of regulated pests. From these plants, first the propagation stocks and then the certified mother plants are obtained according to a hierarchical scheme. Lastly, certified mother plants constitute the stocks from which the certified olive trees for farmers are produced. Propagation of nuclear stocks and of propagation stocks can be performed by seeding or self-rooting for rootstocks, while the varieties can be multiplied by self-rooting or grafting on pathogen-tested rootstocks with the same or a higher degree of certification. The conservation of the different certification materials should take place in insect-proof greenhouses or in the field by recognized official bodies, or in a nursery for the certified mother plants [66]. All these categories of plant materials must be kept in conditions that prevent infection; when their storage takes place in a field, this must be tested for the presence of pests. During all certification steps, the plants are subjected to visual inspections for the symptoms and to diagnostic tests to verify that they are free from the Union-regulated nonquarantine pests of the olive tree and for the priority quarantine pathogen *X. fastidiosa*. However, plants intended for export must be subjected to controls for the pathogens covered by the phytosanitary legislation of the importing country.

## 6. Discussion

Plant protection legislation and phytopathological diagnosis are important tools to minimize the risk of the introduction of invasive plant pests, particularly as a consequence of the frequent outbreaks of non-native species such as the recent epidemic in Southern Italy of olive quick disease syndrome (OQDS) caused by the bacterium *X. fastidiosa* [3]. This disease, in addition to having seriously affected the olive production of the contaminated area [67], also constitutes a serious threat to the wide biodiversity of olive, which has considerable potential that could be exploited for improving the traits of interest of this species [68,69,70,71]. For these reasons, interventions are needed to avoid the loss of these genetic resources [72]. The concept of a plant quarantine is part of the subject areas of plant health and plant protection, and includes national and international legislative measures and the related activities aimed to minimize the risk of the spread of pests that are harmful to the plants in a given region where they are not present, either by preventing their introduction or through eradication and containment in the case of their unintentional entry [6]. These measures may include the prohibition of or restrictions on the exchange of plants or propagation material between countries, or the definition of phytosanitary requirements that commodities must meet in order to be marketed and which should be established following an assessment of the risk posed by the plant pests [4]. Legislative measures also indicated by the term “quarantine” and also include eradication and containment actions, surveys and risk assessments [6], treatment of consignments [4], and inspections by visual observations and laboratory tests to detect the examined pests [73]. Quarantine programs are the first control measure for plants’ protection [74], thus allowing us to preserve the productivity of agricultural crops [75]. Their implementation is an important action in the management of plant diseases and play a significant role in reducing the occurrence of plant pests [76]. The benefits associated with eliminating pathogens in vegetative propagation material have promoted the certification of healthy plant material [77]. Phytosanitary laws are aimed primarily at preventing the introduction of potentially harmful organisms that can pose a serious threat to plant health, which are referred to as quarantine organisms. However, the laws are less restrictive of pests that do not cause significant damage, as they would otherwise lead to unjustified economic disadvantages [78]. Because of the prevalent vegetative propagation of the olive tree over time, this species can be affected by various infectious organisms including viruses, fungi, bacteria, phytoplasma and nematodes, which, in some cases, can seriously affect plants’ health and production. The current phytosanitary legislation constituted by Regulation (EU) 2016/2031 [53] considers some of the harmful organisms for the olive tree which, due to their harmful effects on olive production and plant health, meet the criteria to be considered as Union-regulated non-quarantine pest due to their widespread diffusion in Europe. Their absence is a requirement that olive tree propagation material must have for the issuance of a plant passport or phytosanitary certificate which guarantees its pest-free status. Furthermore, for these harmful agents, the sanitary quality of the olive tree propagation material should be verified for the purpose of certification according to the provisions of EPPO Standard PM 4/17 (3) [64]. This standard also states that certified olive material must undergo specific diagnostic tests for the detection of the quarantine pathogen *X. fastidiosa* as part of the measures to prevent the spread of this bacterium. 

Quarantine pests include non-native organisms that are potentially harmful to the economy and environment of an area if they enter and find a suitable environment, or which, if present, are not widely distributed; in any case, they are subject to interventions for their control. The current European phytosanitary legislation (Regulation (EU) 2016/2031) has introduced a new category of quarantine pests, called Union priority pests due to their high risk, against which greater resources must be addressed by providing for periodic investigations and the implementation of emergency plans by the member states. The bacterium *X. fastidiosa* has been included in this category of quarantine organisms, as specified in the Commission Delegated Regulation (EU) 2019/1702 [59]. Preventing the introduction and spread of high-risk pests is linked to adhesion to the legislative measures and to cooperation among states, which are the objectives pursued by the various authorities for the protection of plants. Furthermore, the participation and training of all operators in the sector also play an important role in guaranteeing the effectiveness of these actions. These measures are often the only and most convenient control measures against plant pathogens, allowing agents not only to protect the agriculture and economy of a region, but also to safeguard natural ecosystems and biodiversity, which are important for the sustainability of agricultural production.

## Figures and Tables

**Figure 1 plants-12-00699-f001:**
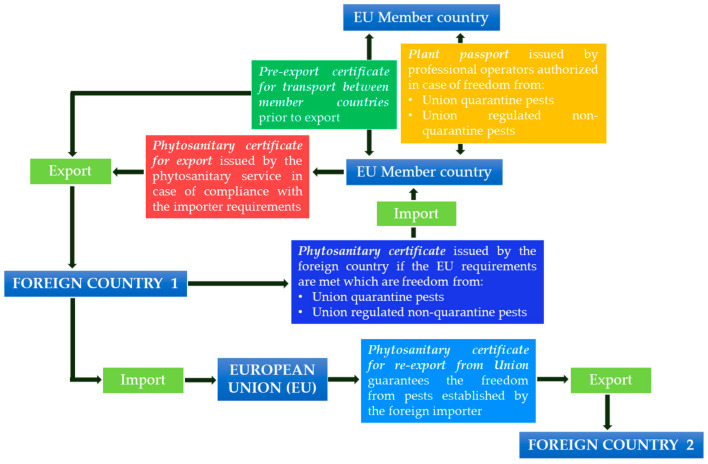
Flowchart representing the steps for the movement of plant propagation materials inside and outside the European community, with reference to the certificates and phytosanitary requirements according to the Community Regulation on Plant Health 2016/2031 [53].

**Table 1 plants-12-00699-t001:** European regulatory framework in force on the prevention of the entry and spread of plant pests in the territory of the European Union.

Regulation (EU) 2016/2031 [53] http://data.europa.eu/eli/reg/2016/2031/oj/eng	Protective measures against plants pests
Commission Implementing Regulation (EU) 2017/2313 [54] http://data.europa.eu/eli/reg_impl/2017/2313/oj/eng	Outlines the specifications of a plant passport for movement within the Union’s territory, and for introduction into and movement within a protected zone
Commission Implementing Regulation (EU) 2018/2018 [55] http://data.europa.eu/eli/reg_impl/2018/2018/oj	Rules concerning the procedure to be followed in order to carry out a risk assessment of high-risk plants and plant products
Commission Implementing Regulation (EU) 2018/2019 [56] http://data.europa.eu/eli/reg_impl/2018/2019/oj/eng	A list of high-risk plants and a list of plants for which no certificates are required for introduction into the Union
Commission Delegated Regulation (EU) 2019/827 [57] http://data.europa.eu/eli/reg_del/2019/827/oj	Criteria to be fulfilled by professional operators authorized to issue plant passports
Commission Implementing Regulation (EU) 2019/2072 [58] http://data.europa.eu/eli/reg_impl/2019/2072/oj/eng	A list of the Union’s quarantine pests (Annex II) and a list of Union-regulated nonquarantine pests (Annex IV)
Commission Delegated Regulation (EU) 2019/1702 [59] http://data.europa.eu/eli/reg_del/2019/1702/oj/eng	A list of priority pests
Commission Delegated Regulation (EU) 2019/829 [60] http://data.europa.eu/eli/reg_del/2019/829/oj	Derogations on the introduction and movement of certain harmful pests or plants within the Union or its protected zones for official trials, scientific or educational purposes, and varietal selection or breeding
Commission Implementing Regulation (EU) 2019/2148 [61] http://data.europa.eu/eli/reg_impl/2019/2148/oj	Rules concerning the release of plants, plant products and other objects from quarantine stations and confinement facilities
Commission Implementing Regulation (EU) 2020/1201 [62] http://data.europa.eu/eli/reg_impl/2020/1201/oj/eng	Measures to prevent the introduction into and the spread within the Union of *Xylella fastidiosa*

## Data Availability

No new data were created or analyzed in this study. Data sharing is not applicable to this article.

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
