# Peer review of "Phytosanitary Rules for the Movement of Olive (Olea europaea L.) Propagation Material into the European Union (EU)"

_plants, 2023, doi:10.3390/plants12040699_

Round 1

Reviewer 1 Report

Dear authors,

I have reviewed your manuscript " Phytosanitary rules for olive (Olea europaea L.) propagation material movement into European Union (EU)" submitted for publication in Plants. The review manuscript is interesting and presents a valuable collection of information on the current plant protection legislation for the exchange of plants or propagation material within the European Union or for export to other countries. Additionally, authors presented a brief summary of the main olive tree pests transmissible with the propagation material.

I have highlighted a few points below that could improve the quality of the manuscript before publication.

Line 37: regarding the absence of specific pests [add a reference to support this sentence].

Line 44: in November 1997 [add a reference to support this sentence].

Line 105: How about viroids?

Line 107: propagation material [add references to support this sentence].

Line 108: olive tree plant [add a reference to support this sentence].

Lines 114-116: Please double check the virus nomenclature. Is it not necessary for the first letter to be capitalized? Check this throughout the manuscript.

Line 130” other plant species [add references to support this sentence].

Line 157: Is this a figure?

Lines 159-etc: Article 3…102 from what? I know what you mean but please make it clear in the text.

Lines 260-262: Describe the virus name only the first time you mention it in the text and then just add the abbreviation

Line 269: pest in the EU [add references to support this sentence].

Lines 305-313: the text looks like a repetition. Consider rewriting this.

Lines 299-320: There are no references to support all these sentence.

Line 322: Exclude extra line.

Line 327: It is not necessary to describe OQDS again.

Lines 347-352: again… it is not necessary to describe what you have already described before.

I suggest adding a flowchart that represents the steps of receiving and distributing propagation materials inside and outside the EU. In addition, describe the importance of quarantine programs for the safe movement of plant material.

Author Response

Response to Reviewer 1 Comments

Dear authors, I have reviewed your manuscript " Phytosanitary rules for olive (Olea europaea L.) propagation material movement into European Union (EU)" submitted for publication in Plants. The review manuscript is interesting and presents a valuable collection of information on the current plant protection legislation for the exchange of plants or propagation material within the European Union or for export to other countries. Additionally, authors presented a brief summary of the main olive tree pests transmissible with the propagation material.

We would like to thank the reviewer for careful reading the manuscript and for making suggestions to improve the manuscript. We have addressed all the points in detail below.

Point 1: Line 37: regarding the absence of specific pests [add a reference to support this sentence].

Response 1: We thank the reviewer for this remark. We have added a reference as suggested.

Point 2: Line 44: in November 1997 [add a reference to support this sentence].

Response 2: We appreciate the suggestion. We have inserted a bibliographic reference to the sentence indicated.

Point 3: Line 105: How about viroids?

Response 3: We thank the reviewer for the observation. Currently, no viroids have been identified for the olive tree, although there are several virus-like diseases whose causal agent is not known. In the work conducted by Herrera and Madariaga in 1999, the aetiology of sickle leaf disease was attributed to a viroid, however Caglayan et al., 2011 reported that this agent was not a viroid. In acceptance of your suggestion, we have integrated into the text inserting the disorders of the olive tree with probable virus-like etiology.

References

-Herrera, M., G., and Madariaga V., M., 1999. Detection of viroid like-organism on olive plants (Olea europaea L.) with sickle leaf symptoms. Agricultura Técnica (Santiago) 59: 178-185.

-Caglayan, K., F. Faggioli, e M. Barba. «CHAPTER 53: Viruses, Phytoplasmas, and Diseases of Unknown Etiology of Olive Trees». In Virus and Virus-Like Diseases of Pome and Stone Fruits, 289–97. Virology. The American Phytopathological Society, 2011. https://doi.org/10.1094/9780890545010.053.

Point 4: Line 107: propagation material [add references to support this sentence].

Response 4: Thanks for this observation. We have added a reference to support the sentence.

Point 5: Line 108: olive tree plant [add a reference to support this sentence].

Response 5: Thanks for the suggestion, we have added a specific reference in the line in question.

Point 6: Lines 114-116: Please double check the virus nomenclature. Is it not necessary for the first letter to be capitalized? Check this throughout the manuscript.

Response 6: Thanks for this remark. We checked the virus nomenclature in the manuscript.

Point 7: Line 130” other plant species [add references to support this sentence].

Response 7: We thank you for your observation. We have added a reference to support this sentence.

Point 8: Line 157: Is this a figure?

Response 8: We thank the reviewer for this remark. We have modified as a table rather than figure the indication of the representation.

Point 9: Lines 159-etc: Article 3…102 from what? I know what you mean but please make it clear in the text.

Response 9: Thanks for this observation. We have added an explanatory sentence to the text which specifies what the articles in brackets in the text refer to.

Point 10: Lines 260-262: Describe the virus name only the first time you mention it in the text and then just add the abbreviation

Response 10: Thank you for this observation. We have made the changes as suggested.

Point 11: Line 269: pest in the EU [add references to support this sentence].

Response 11: Thanks for the suggestion. We have added a bibliographic reference to support the sentence.

Point 12: Lines 305-313: the text looks like a repetition. Consider rewriting this.

Response 12: We would thank reviewer for this remark. We have rewritten this part of the manuscript as advised.

Point 13: Lines 299-320: There are no references to support all these sentence.

Response 13: Thanks for this remark. We have added bibliographic references to support the sentences in question.

Point 14: Line 322: Exclude extra line.

Response 14: Thanks for this observation. We made the indicated change.

Point 15: Line 327: It is not necessary to describe OQDS again.

Response 15: We appreciate the suggestion. We have eliminated this repeated part in the manuscript.

Point 16: Lines 347-352: again… it is not necessary to describe what you have already described before.

Response 16: Thanks for the observation. We have deleted from the manuscript the part present in the indicated lines.

Point 17: I suggest adding a flowchart that represents the steps of receiving and distributing propagation materials inside and outside the EU. In addition, describe the importance of quarantine programs for the safe movement of plant material.

Response 17: We thank the reviewer for the suggestion. We integrated the manuscript by adding a flowchart representing the phases for the import and export of the propagation plant material with particular reference to the necessary phytosanitary certificates and requirements. We have also integrated the discussion section by emphasizing the importance that plant quarantine and all related activities have in guaranteeing to farmers the sanitary quality of plants and plant propagation materials, thus helping to safeguard agricultural production.

Reviewer 2 Report

The document "Phytosanitary rules for olive (Olea europaea L.) propagating material movement into the European Union (EU)" is a document reviewing the current regulations on the introduction of propagating material into the European Union (EU). From my review, no mistakes were found throughout the document.

The authors should review the numbering of the sections. The number 2 is repeated in some sections.

Author Response

Response to Reviewer 2 Comments

The document "Phytosanitary rules for olive (Olea europaea L.) propagating material movement into the European Union (EU)" is a document reviewing the current regulations on the introduction of propagating material into the European Union (EU). From my review, no mistakes were found throughout the document.

We would like to thank the reviewer for reading the manuscript carefully and for comments. We have addressed the point in detail below.

Point 1: The authors should review the numbering of the sections. The number 2 is repeated in some sections.

Response 1: We thank for this observation, we made the suggested correction by changing the section numbering.
